# Mature B cells and mesenchymal stem cells control emergency myelopoiesis

Vivian Y Lim[1],*, Xing Feng[1],*, Runfeng Miao[1], Sandra Zehentmeier[1], Nathan Ewing-Crystal[1], Moonyoung Lee[2], Alexei V Tumanov[3], Ji Eun Oh[1], Akiko Iwasaki[1,4], Andrew Wang[1,5], Jungmin Choi[2,6], João P Pereira[1]

**Systemic inflammation halts lymphopoiesis and prioritizes myeloid cell production. How blood cell production switches from homeostasis to emergency myelopoiesis is incompletely understood. Here, we show that lymphotoxin-*β* receptor (LT*β*R) signaling in combination with TNF and IL-1 receptor signaling in bone marrow mesenchymal stem cells (MSCs) down-regulates *Il7* expression to shut down lymphopoiesis during systemic inflammation. LT*β*R signaling in MSCs also promoted CCL2 production during systemic inflammation. Pharmacological or genetic blocking of LT*β*R signaling in MSCs partially enabled lymphopoiesis and reduced monocyte numbers in the spleen during systemic inflammation, which correlated with reduced survival during systemic bacterial and viral infections. Interestingly, lymphotoxin-*α*1*β*2 delivered by B-lineage cells, and specifically by mature B cells, contributed to promote *Il7* down-regulation and reduce MSC lymphopoietic activity. Our studies revealed an unexpected role of LT*β*R signaling in MSCs and identified recirculating mature B cells as an important regulator of emergency myelopoiesis.**

## Introduction

Blood cell production is a tightly regulated process that is critical for organismal homeostasis. Myeloid, erythroid, and lymphoid lineages are continuously produced at defined rates under homeostasis, but during infection and/or systemic inflammation, the production of short-lived innate immune cells of myeloid lineage is prioritized, whereas lymphopoiesis is temporarily shut down, a process described as emergency myelopoiesis (Ueda et al., 2004, 2005; Manz & Boettcher, 2014). This response is essential for survival during infections as myeloid cells are required for pathogen clearance and for instructing adaptive immunity. A large body of

work has shown that pattern recognition receptors (PRRs) expressed on hematopoietic stem and progenitor cells have an important role in skewing hematopoietic differentiation toward myeloid lineages (Boettcher & Manz, 2017; Capitano, 2019; Schultze et al, 2019). However, the mechanisms controlling the shutdown of lymphopoiesis remain poorly understood.

Mesenchymal stem cells (MSCs) and endothelial cells in the bone marrow form a specialized niche for hematopoietic stem cell (HSC) maintenance and differentiation into lymphoid lineages because of their expression of critical stem cell maintenance and lymphopoietic cytokines, namely, KITL and IL-7 (Mendez-Ferrer et al, 2010; Ding et al, 2012; Cordeiro Gomes et al, 2016; Fistonich et al, 2018; Comazzetto et al, 2019). Recent single-cell RNA-sequencing studies revealed that in addition to stem cell maintenance and lympho-poietic cytokines, individual MSCs also express additional lymphoid and myeloid cell lineage–instructive cytokines (Baryawno et al, 2019; Tikhonova et al, 2019), suggesting that myeloid cell production could also be controlled by the MSC niche. In support of this model, myeloid progenitors have been shown to localize in proximity to MSCs under homeostasis and during inflammation/infection (Herault et al, 2017; Zhang et al, 2021). This model of centralized control of hematopoietic stem and progenitor cell differentiation at the bone marrow MSC niche (Miao et al, 2020) led us to hypothesize that MSCs are able to sense and respond to systemic cues such as inflammatory cytokines to coordinate HSC differentiation for an appropriate hematopoietic response.

## Results

### Systemic inflammation turns off lymphopoietic niche activity by blocking IL-7 production

To better understand the effects of systemic inflammation on HSC differentiation, we treated mice with CFA i.p. for several days and quantified hematopoietic stem and progenitor cell subsets within

---

[1]Department of Immunobiology, School of Medicine, Yale University, New Haven, CT, USA  [2]Department of Biomedical Sciences, Korea University College of Medicine, Seoul, South Korea  [3]Department of Microbiology, Immunology and Molecular Genetics, University of Texas Health Science Center at San Antonio, San Antonio, TX, USA  [4]Howard Hughes Medical Institute, Chevy Chase, MD, USA  [5]Department of Medicine (Rheumatology), School of Medicine, Yale University, New Haven, CT, USA  [6]Department of Genetics, School of Medicine, Yale University, New Haven, CT, USA

Correspondence: joao.pereira@yale.edu; jungminchoi@korea.ac.kr
*Vivian Y Lim and Xing Feng contributed equally to this work

---

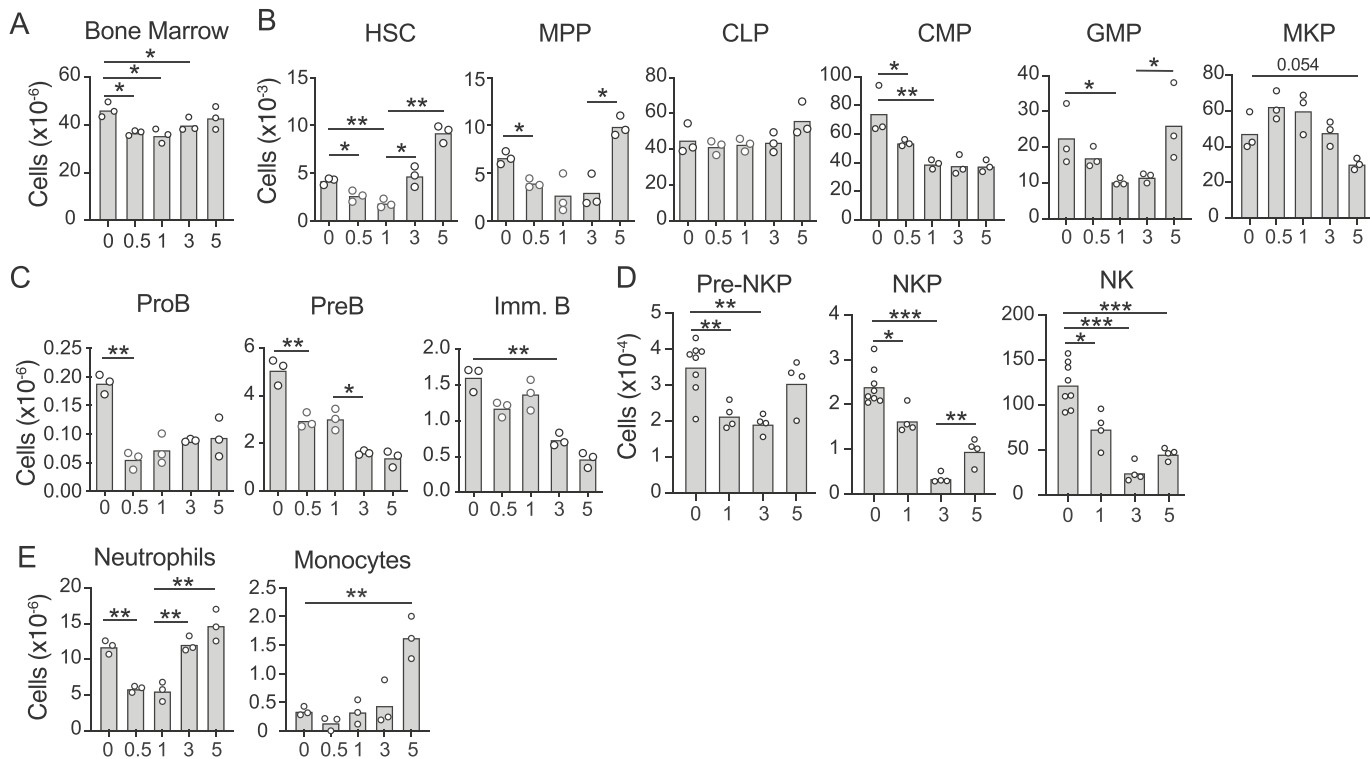

**Figure 1. Effects of systemic inflammation on hematopoiesis.**
**(A)** Total cell number in bone marrow from femur and tibia. **(B)** HSC, MPP, CLP, CMP, GMP, and MKP numbers in bone marrow. **(C)** ProB-, PreB-, and immature B-cell numbers in bone marrow. **(D)** pre-NKP, NKP, and NK cell numbers. **(E)** Bone marrow neutrophil and monocyte numbers in femur and tibia. Mice were administered CFA i.p. on day 0 and analyzed at the indicated time points (days, x-axis). Bars indicate the mean, and circles depict individual mice. *$P < 0.05$ and **$P < 0.005$ by a $t$ test. Data are representative of three independent experiments.

the bone marrow by flow cytometry. The total bone marrow cellularity reduced within the first 12 h and remained lower than that during homeostasis until day 3 (Fig 1A). HSCs and multipotent progenitor cells (MPPs) were numerically reduced at 12 and 24 h after CFA administration, but their numbers increased between days 1 and 5 (Fig 1B). In contrast, common lymphoid progenitor and megakaryocyte progenitor numbers remained unchanged, whereas common myeloid progenitors (CMPs) and granulocyte and monocyte progenitors (GMPs) declined until day 3, with GMPs recovering on day 5. Within the lymphoid lineages, proB-cell numbers dropped sharply 12 h post-CFA administration and remained low through day 5. PreB- and immature B-cell numbers changed more gradually, declining progressively from 12 h through day 5 (Fig 1C). NK cells and NK progenitors were also significantly reduced within 12 h of systemic inflammation, and numbers remained lower than homeostatic levels until day 5 (Fig 1D). Neutrophils and monocytes were also reduced at 12 h, but both cell populations expanded to reach numbers higher than baseline by day 5 post-CFA (Fig 1E).

Because the shutdown in lymphopoiesis was most evident at the IL-7–dependent proB-cell stage, we analyzed the effect of CFA administration on IL7 expression in bone marrow MSCs using an *Il7*-GFP reporter mouse strain (Miller et al, 2013). Indeed, CFA induced a profound down-regulation of *Il7*-GFP in leptin receptor (Lepr)+ MSCs 5 d after administration (Fig 2A and B). The number of Lepr+ MSCs remained unchanged (Fig 2C), suggesting that IL-7 is reduced during systemic inflammation. Consistent with this possibility, IL-7

receptor abundance, which is controlled by an IL-7 receptor–mediated feedforward loop (Ochiai et al, 2012; Clark et al, 2014), declined in proB and preB cells during systemic inflammation (Fig S1A). To test whether reduced IL-7 availability during systemic inflammation caused the shutdown in lymphopoiesis, we treated mice with recombinant IL-7 complexed with a neutralizing anti-IL-7 (aIL-7, clone M25) monoclonal antibody, which increases the half-life of recombinant IL-7 in vivo (Boyman et al, 2008). In vivo IL-7/aIL-7 treatment raised the number of proB and preB cells significantly under non-inflammatory conditions (Fig S1B). During CFA-induced systemic inflammation, IL-7/aIL-7 treatment restored proB- and preB-cell numbers in the bone marrow but did not affect immature and mature B cells, which are not dependent on IL-7 (Fig 2D). Interestingly, IL-7/aIL-7 treatment also blunted the expansion of bone marrow myeloid populations (Fig 2E). Altogether, these studies suggest that inflammation-induced lymphopenia is at least partly controlled by reduced IL-7 production in bone marrow MSCs.

## LTβR signaling and inflammatory cytokine receptor signaling control *Il7* transcription in MSCs

Prior studies have shown critical roles of inflammatory cytokines in driving emergency myelopoiesis (Cain et al, 2009; Manz & Boettcher, 2014). To identify the signals and receptors on MSCs responsible for controlling IL7 transcription in response to inflammation, we performed RNA-sequencing of Lepr+ bone marrow MSCs. Transcriptomic

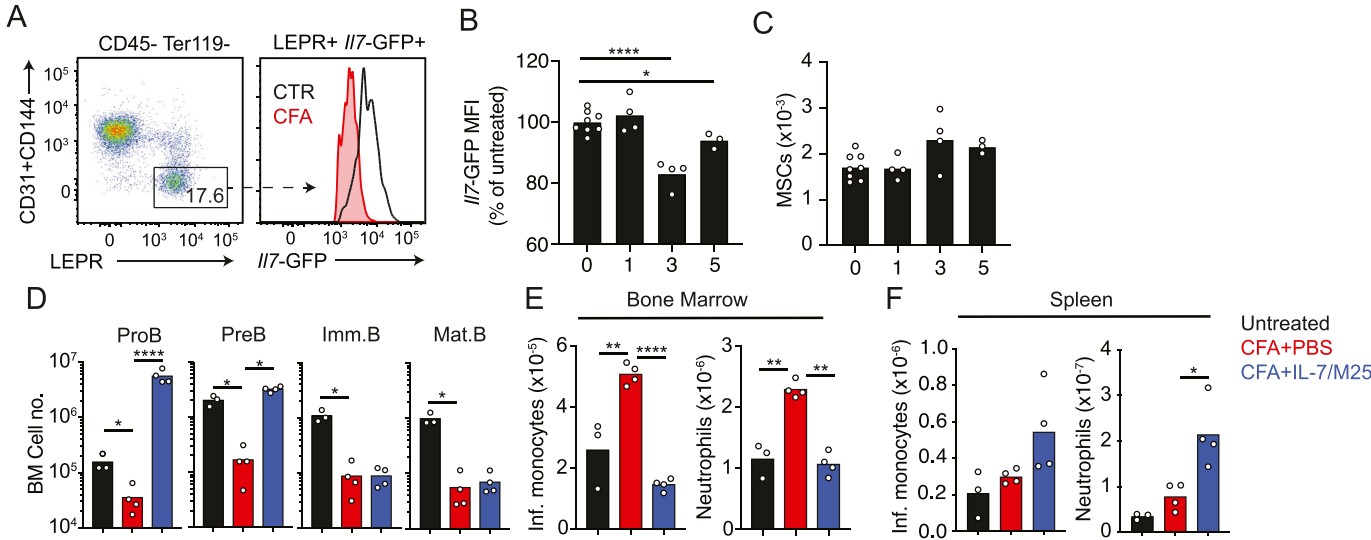

**Figure 2. Inflammation-induced *Il7* down-regulation and impact on emergency myelopoiesis.**
**(A)** *Il7*-GFP expression on gated bone marrow MSCs in mice injected i.p. with CFA (red) or saline (blue) for 5 d. **(B)** *Il7*-GFP Geo. mean in gated Lepr+ MSCs. **(C)** Number of Lepr+ MSCs in bone marrow. **(D, E, F)** Mice immunized with CFA i.p. and pretreated with the IL-7/aIL-7 cytokine/antibody complex (1.5/15 µg, respectively, clone M25) i.v. **(D)** Developing B-cell subsets in bone marrow. **(E, F)** Myeloid cell subsets in bone marrow (E) and spleen (F). Bars represent the average, and circles depict individual mice. *P < 0.05; **P < 0.005; ***P < 0.0005; and ****P < 0.00005 by an unpaired *t* test. Data in all panels are representative of two experiments.

analyses showed that Lepr+ MSCs express multiple inflammatory cytokine receptors, including lymphotoxin-β receptor (LTβR; Fig 3A), the latter being a TNF superfamily receptor that is required for peripheral lymphoid organ development, and for maturation of stromal cell subsets in secondary lymphoid organs (Ruddle, 2014; Cheng et al, 2019). MSCs also expressed low to undetectable amounts of PRRs (Fig 3A), but a prior study showed no measurable roles of PRRs in MSCs during emergency myelopoiesis (Boettcher et al, 2014). To test whether inflammatory cytokines and/or LTβR control IL7 expression in bone marrow MSCs, we challenged *Il7^GFP/+* mice with CFA for 5 d and blocked LTβR and/or inflammatory cytokines using a soluble LTβR-Ig decoy and cytokine blocking antibodies. Interestingly, inflammation-induced IL7 down-regulation was partially blocked with LTβR-Ig, anti-IL-1β, or anti-TNF-α, and was completely blocked with LTβR-Ig + anti-TNF-α (Fig 3B). Although IFN-γ has been shown to play important roles in emergency myelopoiesis (Baldridge et al., 2010; de Bruin et al., 2012; Kaufmann et al, 2018), blocking IFN-γ did not prevent IL7 down-regulation (Fig 3B), nor did it recover B-cell progenitor numbers in vivo (Fig S1C). The complete block of IL7 down-regulation with LTβR-Ig and anti-TNF-α treatment led to significantly increased IL7–dependent progenitor B-cell production, which resulted in increased numbers of immature B cells (Fig 3C). Nevertheless, proB- and preB-cell numbers were still reduced when compared to baseline (Fig 3C), likely because of down-regulation of CXCL12 (Fig S1D). Systemic inflammation and infection down-regulate CXCL12 transcripts and protein levels in vivo (Ueda et al., 2004, 2005; Saw et al, 2019), whereas reduced CXCR4 function (CXCL12 receptor) in B-cell progenitors impairs B lymphopoiesis (Beck et al, 2014; Cordeiro Gomes et al, 2016; Mandal et al, 2019). Furthermore, IL-1 has been reported to directly inhibit the differentiation of MPPs to the B lineage (Kennedy & Knight, 2015) and triple blocking of LTβR, IL-1R, and TNFR signaling resulted in a small but significant increase in B-lineage

cell production when compared to LTβR and TNFR double blocking (Fig S1E). Likewise, combined blocking of IL-1β and TNF-α also completely restored proB-cell numbers in the BM (Fig S1E), whereas LTβR and TNFR double blocking only partially restored proB-cell numbers (Fig 3C). These results show that redundancy between IL-1R, TNFR, and LTβR signaling promotes IL7 down-regulation.

Monocytes and neutrophils were significantly increased in CFA-inflamed mice but were modestly increased in the bone marrow of LTβR-Ig–treated when compared to Hel-Ig–treated mice (Fig 3D). In contrast, both myeloid cell subsets were significantly reduced in the spleen of mice treated with LTβR-Ig, and even further reduced in mice treated with combined LTβR-Ig and anti-TNF-α antibodies (Fig 3E). The concomitant accumulation in bone marrow and reduction in the periphery suggested that myeloid cell egress from bone marrow was impaired by blocking LTβR or TNFR signaling. LTβR and TNFR signaling blockade also reduced the rate of myeloid progenitor proliferation in vivo, whereas BrdU incorporation in common lymphoid progenitors and in megakaryocyte/erythroid progenitors (MEPs) was unaffected in vivo (Fig S1G). In line with the reported role of IL-1R in supporting inflammation-induced emergency myelopoiesis (Ueda et al, 2009) and LTβR signaling in the homeostasis of neutrophils in the spleen (Shou et al, 2021), blocking of IL-1R in addition to LTβR and TNFR led to a further reduction in inflammatory monocytes and neutrophil numbers in the spleen (Fig S1F).

As the role of LTβR signaling in bone marrow MSCs in the control of emergency myelopoiesis was unexpected, we decided to focus our attention on this signaling axis. To understand which MSC-expressed genes are controlled by LTβR signaling, we examined global transcriptomic changes in MSCs isolated from CFA-injected mice that were also pretreated with LTβR-Ig or with control IgG1 (anti-hen egg lysozyme, HEL-Ig) for 1 d. Systemic inflammation segregated MSCs from untreated samples on principal component

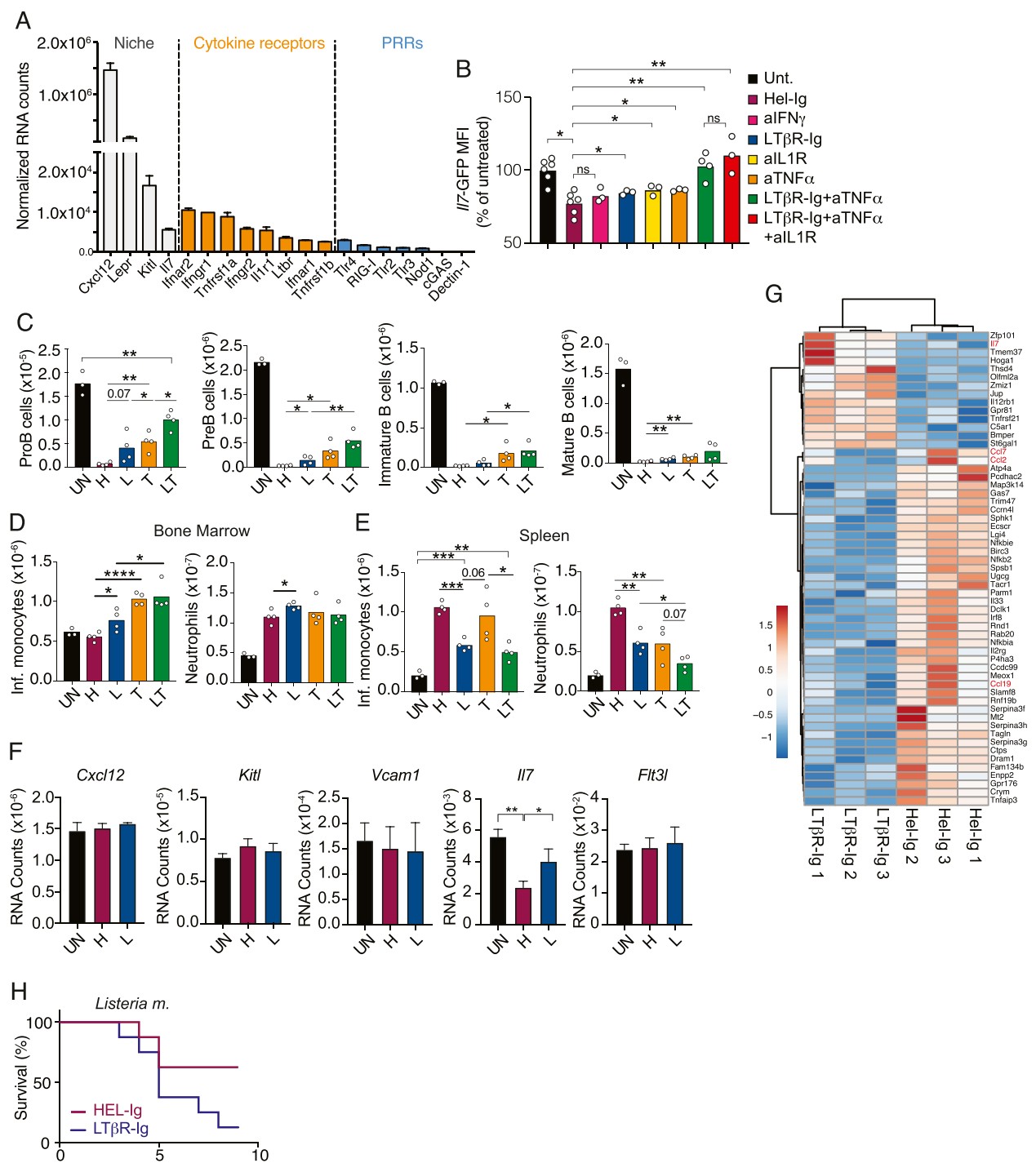

**Figure 3. LT*β*R signaling and TNF receptor signaling control IL-7 production and B lymphopoiesis.**
**(A)** Normalized RNA counts of cytokine and pattern recognition receptors in bone marrow MSCs; data obtained from MSC RNA sequencing. **(B)** *Il7*-GFP expression in gated bone marrow MSCs relative to untreated mice (expressed as percent of untreated). **(C)** Numbers of proB, preB, immature B, and mature B cells in bone marrow. **(D, E)** Numbers of inflammatory monocytes and neutrophils in bone marrow (D) and spleen (E). In panels (B, C, D, E), data were obtained from untreated control (UN, black) or CFA-immunized mice treated with HEL-Ig (H, wine red), LT*β*R-Ig (L, blue), anti-TNF-*α* (T, orange), and LT*β*R-Ig + anti-TNF-*α* (LT, green) for 5 d. **(F)** CFA-induced changes in lymphopoietic factors expressed by MSCs. **(G)** Differentially expressed genes in MSCs isolated from mice injected with CFA for 24 h and treated with LT*β*R-Ig (L) or HEL-Ig (H, FDR-adjusted q-value < 0.05). **(H)** Survival after i.v. infection with *Listeria monocytogenes* (50,000 CFU) after pretreatment with either HEL-Ig or LT*β*R-Ig (n = 8 per group). Data in panels (A, B, C, D, E) are representative of two to six experiments. Data in panels (F, G) were from one experiment. Bars represent the average, circles depict individual mice, and error bars indicate the SEM. *P < 0.05; **P < 0.005; ***P < 0.0005; and ****P < 0.00005 by an unpaired *t* test.

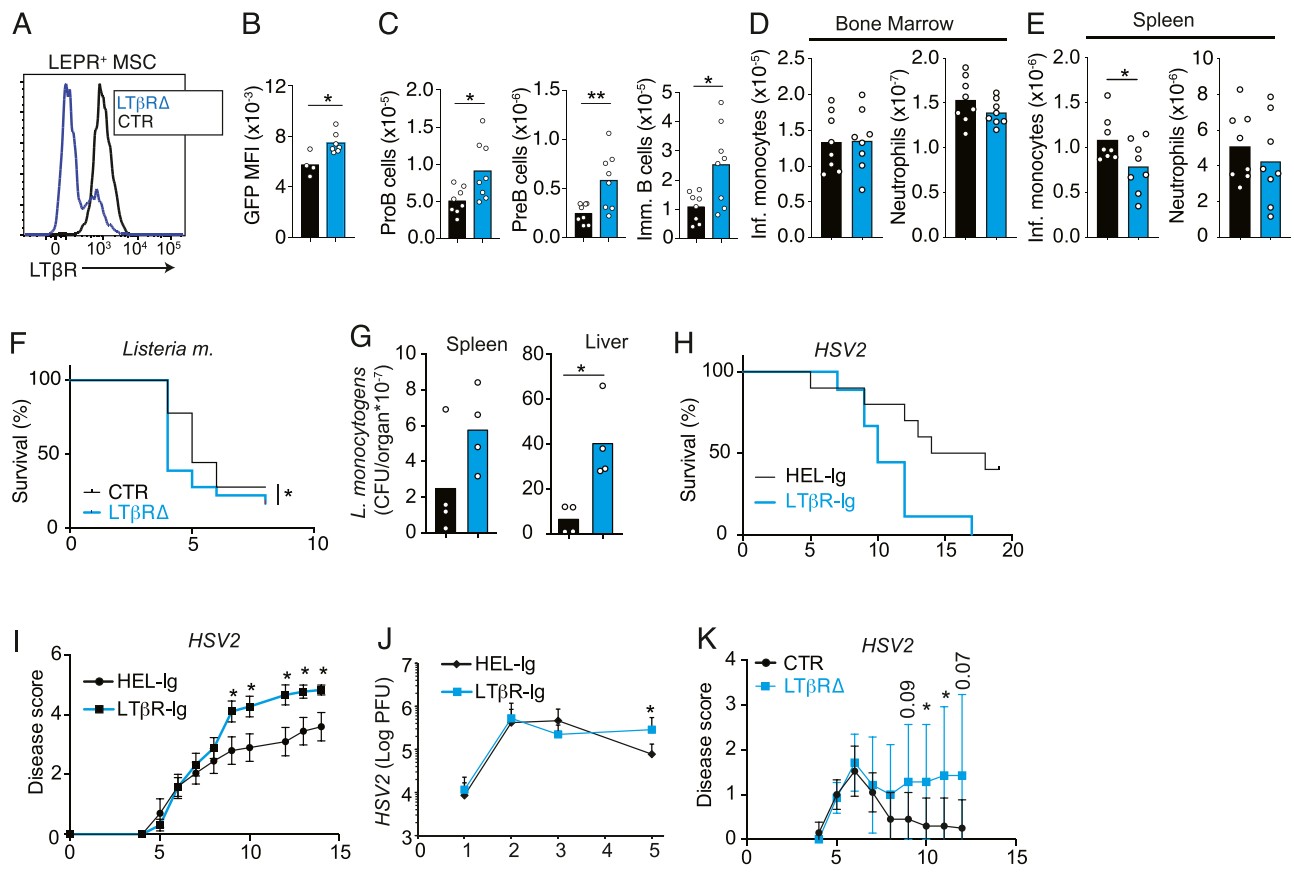

**Figure 4. LTβR signaling in bone marrow MSCs controls emergency myelopoiesis.**
**(A)** Histogram of LTβR expression on gated bone marrow Lepr+ MSCs from *Lepr⁺/⁺; Ltbr^{fl/fl}* (CTR, black) and *Lepr^{Cre/+}; Ltbr^{fl/fl}* (blue, LTβRΔ) mice. **(B, C, D, E)** *Il7* expression in MSCs, (C) developing B-cell subsets in bone marrow, and (D, E) inflammatory monocytes and neutrophils in bone marrow (D) and in the spleen (E) of CTR (black) and of LTβRΔ (blue) mice immunized with CFA i.p. for 5 d. Bars represent the average, circles depict individual mice, and error bars indicate the SEM. **(F)** Survival of CTR (black, n = 9) and of LTβRΔ (blue, n = 11) mice after i.v. infection with *Listeria monocytogenes* (370,000 CFU). **(G)** *Listeria monocytogenes* CFUs in spleen and liver of CTR (black) and of LTβRΔ (blue) mice at day 3 post-infection. **(H, I, J)** Survival (H), disease score (I), and HSV-2 viral titers (J) in vaginal wash of mice infected with 1,000 PFU after pretreatment with either HEL-Ig (n = 8) or LTβR-Ig (n = 6). **(K)** Disease score of CTR (black) and of LTβRΔ (blue) mice infected with HSV2 (1,000 PFU) intravaginally. In panels (G, H, I, J), x-axis shows time post-infection (days). *P < 0.05 and **P < 0.005 by an unpaired *t* test. Data in panels (A, B, C, D, E, F) are representative of two to four experiments. Data in panels (G, H, I, J) were from one experiment. Bars represent the average, circles depict individual mice, and error bars indicate the SEM.

(PC)-1, but LTβR-Ig treatment did not separate samples on PC1 or PC2 (Fig S1I). CFA significantly changed the expression of 576 genes, of which 56 were directly controlled by LTβR signaling (Table S1). Gene set enrichment analyses revealed up-regulation of genes associated with metabolic changes (e.g., oxidative phosphorylation, reactive oxygen species, and Myc-regulated genes) and with increased inflammatory cytokine receptor signaling (e.g., NF-κB and STAT5 signaling) in MSCs isolated from CFA-injected and Hel-Ig mice. LTβR blocking led to increased expression of genes associated with mesenchymal cell differentiation and angiogenesis, and reduced expression of genes associated with inflammation (Fig S1J). Importantly, among factors important for lymphopoiesis, such as CXCL12, Kit and FLT3 ligands, VCAM1, and IL7, CFA-induced inflammation only promoted IL7 down-regulation, which was partially prevented by LTβR blocking (Fig 3F). Besides IL7, LTβR signaling presumably in MSCs also controlled CCL2 expression (Figs 3G and S1H and Table S1), consistent with prior studies (Cuff et al, 1999). CCL2 is a chemokine that promotes monocyte egress from bone marrow by binding to its receptor CCR2 (Serbina & Pamer, 2006; Shi

et al, 2011). CCL7, another ligand for CCR2, was also altered by LTβR blocking during systemic inflammation (Fig 3G). In contrast, LTβR-Ig did not affect CXCL12 transcription (Figs 3F and S1D and Table S1). Because CCR2 signaling promotes CXCR4 desensitization in monocytes (Jung et al, 2015), these data could explain the apparent defect in myeloid cell egress from bone marrow during systemic inflammation with LTβR and TNFR blocking (Fig 3D and E). Although the LTβR-dependent fold changes observed in MSC gene expression, and in lymphoid and myeloid cell production, were relatively small (<threefold), these were nevertheless physiologically important given that mice became more susceptible to systemic infection with *Listeria monocytogenes* (Fig 3H), which is consistent with a prominent role of monocytes in defense against systemic *L. monocytogenes* infection (Serbina et al, 2008).

To determine whether cell-intrinsic LTβR signaling in MSCs is required for emergency myelopoiesis, we challenged mice conditionally deficient in *Ltbr* in bone marrow MSCs (LTβR cKO) with CFA. *Lepr*-cre–mediated deletion abrogated LTβR with about 80% efficiency in bone marrow MSCs (Fig 4A), which was sufficient for

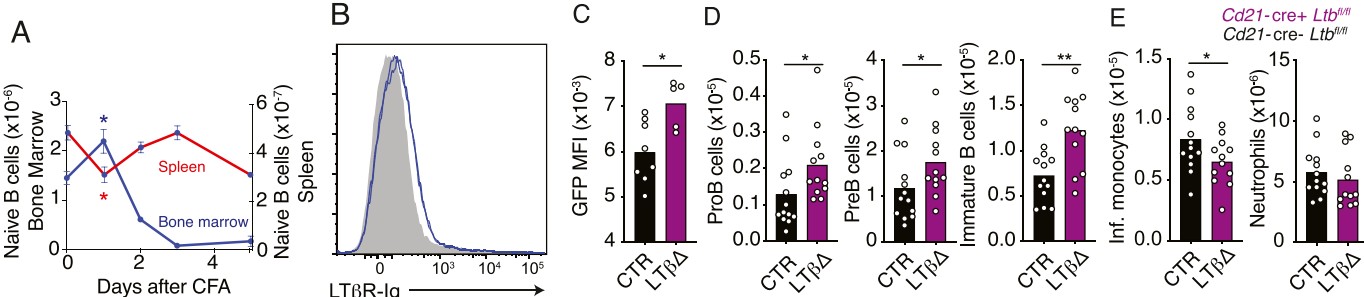

**Figure 5. Circulatory mature B cells provide lymphotoxin-α1β2 to MSCs and promote emergency myelopoiesis.**
**(A)** Changes in mature B-cell number in bone marrow (blue) and spleen (red) of mice injected with CFA (i.p.) over 5 d. **(B)** LTβR ligand expression in mature B cells in bone marrow of mice immunized with CFA for 24 h ($Ltb^{-/-}$, gray-filled histogram; $Ltb^{+/+}$, blue histogram). **(C, D, E)** *Il7*-GFP expression Geo. mean in MSCs (C); developing B-cell subsets in bone marrow (D); and inflammatory monocytes and neutrophils in the spleen (E) of control (black) or *Cd21*-cre $Ltb^{fl/fl}$ mice (purple) injected with CFA i.p. for 5 d. Bars represent the average, and error bars indicate the SEM. *$P < 0.05$ and **$P < 0.005$ by an unpaired $t$ test. Data in all panels are representative of two to four experiments.

preventing inflammation-induced IL7 down-regulation (Fig 4B). This resulted in increased B-cell progenitor production during systemic inflammation (Fig 4C). Although monocyte and neutrophil numbers in the bone marrow were similar, splenic monocyte numbers were significantly reduced (Fig 4D and E). The difference in bone marrow monocyte numbers between systemic LTβR-Ig treatment and LTβR cKO mice during inflammation may be due to the fact that endothelial cells in bone marrow, which express CCL2 and contribute to monocyte egress (Shi et al, 2011), also express LTβR (Tikhonova et al, 2019).

Conditional *Ltbr* deficiency in MSCs did not alter the rate of hematopoietic progenitor, monocyte, and neutrophil production, as measured by BrdU incorporation over a 2-h period (Fig S2A). However, there was a trend toward reduced numbers of newly generated BrdU+ monocytes reaching the spleen over a period of 48-h of BrdU exposure (Fig S2B), and a significant reduction in the total number of monocytes in the spleen (Fig S2C). During homeostasis, LTβR cKO mice had a higher number of proB cells in the bone marrow, consistent with our prior studies (Zehentmeier et al, 2022), but similar numbers of monocytes and neutrophils in bone marrow and spleen (Fig S2E–G). The number of MSCs was unchanged (Fig S2D). Although *Ltbr* deficiency in MSCs led to smaller than twofold changes in B-cell and myeloid cell numbers during inflammation, these changes were physiologically relevant given that LTβR cKO mice were significantly more susceptible to systemic infection with *L. monocytogenes* by day 5 (Fig 4F). Consistent with this, proB-cell numbers were increased in the bone marrow of LTβR cKO mice infected with *Listeria* on day 2 (Fig S2H–J), and splenic monocytes were significantly reduced on day 3 (Fig S2K–M). Importantly, LTβR cKO animals had significantly higher pathogen burden compared with WT mice (Fig 4G), suggesting that the impairment in emergency myelopoiesis compromised pathogen clearance. However, we cannot rule out the possibility that alterations in splenic niches of LTβR cKO mice may have contributed to the reduction in splenic monocytes given the fact that a few stromal cells are targeted by Lepr-cre (Zehentmeier et al, 2022). To test whether the LTβR axis was important in systemic infectious inflammation generally, we also examined models of acute viral challenges. As in the CFA model of bacterial inflammation, challenge with poly-inosinic:polycytidylic acid (polyI:C), a mimetic of viral RNA, also reduced IL7 expression and B-cell production, and promoted

emergency myelopoiesis in an LTβR (and TNF-α)-dependent manner (Fig S3A–D). These findings are consistent with early studies showing a requirement for TNF-α and lymphotoxin-α in B-cell progenitor apoptosis induced by systemic infections with influenza A virus (Sedger et al, 2002). In contrast, conditional *Ifnar1* deficiency in MSCs did not impact IL7 expression nor B-cell and myeloid cell development during polyI:C-induced inflammation (Fig S3E–G). Like in the *Listeria* model, mice treated with LTβR-Ig, or mice conditionally deficient in *Ltbr* in bone marrow MSCs, were significantly more susceptible to infection of HSV2 (Fig 4H–K), a pathogen that is also controlled by monocyte-mediated immunity (Iijima et al, 2011).

Systemic inflammatory cytokines, such as IL-1 and TNF-α, act on the hypothalamic–pituitary–adrenal axis to increase glucocorticoid (GC) production, which in turn exerts anti-inflammatory effects (Cain & Cidlowski, 2017). GCs also act directly on B and T lymphocytes by activating the mitochondrial pathway of cellular apoptosis (Cain et al, 2020; Taves & Ashwell, 2021). However, conditional deletion of the GC receptor NR3C1 in B-lineage cells had no effect on CFA-induced lymphopoiesis shutdown and emergency myelopoiesis (Fig S3H and I). Furthermore, lymphopoiesis and emergency myelopoiesis were indistinguishable in *Nr3c1* cKO and control littermate mice treated with LTβR and TNF-α blocking reagents (Fig S3J and K), ruling out a significant role of GCs in emergency myelopoiesis.

### Lymphotoxin-α1β2 from recirculating mature B cells promotes emergency myelopoiesis

Finally, we focused on determining the source of LTβR ligands. In the first 24 h after systemic infection, although myeloid cells and developing B cells are significantly reduced (Fig 1), mature B cells accumulate in the bone marrow and are reduced in the spleen (Fig 5A), consistent with prior observations (Moreau et al, 2015). Thus, we hypothesized that recirculating mature B cells might play a role in driving IL7 down-regulation given the fact that naïve B cells express lymphotoxin-α1β2 heterotrimers (a membrane-bound ligand for LTβR) and organize the stromal compartments of secondary lymphoid organ through interactions with stromal cell–expressed LTβR (Fu & Chaplin, 1999; Cyster, 2005). Furthermore, naïve B cells transiting through the bone marrow express measurable amounts of LTβR ligands even during systemic inflammation (Fig 5B), which

led us to consider the possibility that these cells deliver lymphotoxin-$\alpha$1$\beta$2 signals to LT$\beta$R on bone marrow MSCs during the early hours of systemic inflammation. Importantly, MSCs expressed significantly higher IL7 transcripts during systemic inflammation when naïve B cells were conditionally deficient in *Ltb* using *Cd21*-cre (Fig 5C). Similar results were obtained when B-lineage cells were conditionally deficient in *Ltb* (Fig S3L–N). In contrast, no changes in *Il7*-GFP, developing B-cell subsets, or myeloid cells were noted in the bone marrow and spleen of mice conditionally deficient in *Ltb* in B-lineage cells during homeostasis (Fig S3O–Q). Increased IL7 was consistent with increased B-cell production in bone marrow, which correlated with significantly reduced monocytes in the spleen (Fig 5D and E), similar to that seen in LT$\beta$R cKO (Fig 4D). These data revealed an unexpected and previously unknown role of naïve recirculating B cells in turning off B-lymphopoiesis through IL7 down-regulation in bone marrow MSCs during systemic inflammation. It also suggests that by engaging LT$\beta$R on MSCs, mature B cells contribute to promoting monocyte egress from bone marrow and fine-tuning emergency myelopoiesis. Hematopoietic stem and progenitor cells, and myeloid cells, in the bone marrow also produce LT$\beta$R ligands (Zehentmeier et al, 2022). Although it is possible that other hematopoietic cells deliver LT$\beta$R ligands to MSCs, our results suggest that such contribution may be minimal.

## Discussion

In this study, we showed that LT$\beta$R signaling, IL-1R signaling, and TNF-$\alpha$ receptor signaling in bone marrow MSCs promote lymphopoiesis shutdown by turning off IL7 production. Mature recirculating B cells accumulate in the bone marrow for a period of 12–24 h during systemic inflammation where they deliver LT$\alpha$1$\beta$2 to LT$\beta$R expressed on MSCs. Besides shunting IL7 production, LT$\beta$R signaling also modulates the expression of CCL2, a chemokine expressed by MSCs that engages CCR2 and promotes monocyte egress from bone marrow (Serbina & Pamer, 2006; Shi et al, 2011).

The reciprocal production of lymphocytes and myeloid cells during systemic infection suggests that expansion of myelopoiesis cannot occur, whereas homeostatic numbers of lymphoid cells are still being produced. By turning off IL7 transcripts and promoting myeloid cell egress, LT$\beta$R, in combination with TNFR and IL-1R signaling in MSCs, acts as a molecular switch between homeostatic hematopoiesis and a temporary state of lymphopenia that is required for emergency myelopoiesis to proceed. Why lymphopoiesis needs to be turned off temporarily to allow myeloid cell expansion remains unknown. We speculate that sickness behaviors, and particularly anorexia induced by systemic infection (a metabolic adaptation that is essential for organismal tolerance to systemic inflammation (Wang et al, 2016)), impose metabolic constraints in the bone marrow microenvironment possibly because of reduced cytokines (e.g., SCF) or nutrient availability (e.g., glucose, amino acids) that may enforce an upper limit to the rate of myeloid cell production. It is possible that some blood cancers, such as Philadelphia chromosome acute lymphoblastic leukemia and acute myeloid leukemias, which are known to induce IL7 down-regulation in bone marrow MSCs (Fistonich et al, 2018; Baryawno et al, 2019; Zehentmeier & Pereira, 2019) exploit

molecular mechanisms, like LT$\beta$R, to prevent normal blood cell development and in this way reduce competition for limiting nutrients in the bone marrow interstitium.

## Materials and Methods

### Mice

C57BL/6 mice were purchased from the Jackson Laboratories or the National Cancer Institute. *Lepr*-cre, *Mb1*$^{cre/+}$, *Nr3c1*$^{fl/fl}$, *Ifnar*$^{fl/fl}$, and *Cd21*-cre mice were purchased from the Jackson Laboratories. *Il7*$^{GFP/+}$ mice were from our internal colony. *Ltb*$^{fl/fl}$ (Tumanov et al, 2002) and *Ltbr*$^{fl/fl}$ (Wang et al, 2010) mice were bred at Yale Animal Resources Center. Adult males (8–13 wk) were used for *L. monocytogenes* infections, and adult females (8–12 wk) were used for HSV-2 infections. Male and female adult mice (8–12 wk) were used for all other experiments. All mice were maintained under specific pathogen-free conditions at the Yale Animal Resources Center and were used according to the protocol approved by the Yale University Institutional Animal Care and Use Committee.

### Flow cytometry

BM stromal cells were isolated as previously described (Cordeiro Gomes et al, 2016). Briefly, long bones were flushed with HBSS supplemented with 2% of heat-inactivated fetal bovine serum, penicillin/streptomycin, L-glutamine, Hepes, and 200 U/ml collagenase IV (Worthington Biochemical Corporation), and digested for 30 min at 37°C. Cell clumps were dissociated by gentle pipetting. Cells were filtered through a 100-$\mu$m nylon mesh and washed with HBSS/2% FBS. All centrifugation steps were done at 300$g$ for 5 min, and all stains were done on ice. LEPR stains were done for 1 h and all other stains for 20 min. BM MSCs were identified as CD45$^-$ Ter119$^-$ CD31$^-$ CD144$^-$ LEPR$^+$ cells. For analysis of hematopoietic populations, long bones were flushed with DMEM supplemented with 1.5% fetal bovine serum, penicillin/streptomycin, L-glutamine, and Hepes and spleens were mashed through a 70-$\mu$m nylon mesh. Red blood cells were lysed with ammonium chloride buffer. Hematopoietic cell populations were identified as follows: ProB: CD19$^+$ CD93$^+$ IgM$^-$ cKit$^+$; PreB: CD19$^+$ CD93$^+$ IgM$^-$ cKit$^-$; immature B: CD19$^+$ IgM$^+$ CD93$^+$; mature B: CD19$^+$ IgM$^+$ CD93$^-$; developing neutrophils: CD115$^-$ Gr1$^+$ CD11b$^{hi}$ CXCR4$^{hi}$; mature neutrophils: CD115$^-$ Gr1$^{hi}$ CD11b$^{lo}$; immature monocytes: CD115$^+$ Gr1$^+$ CXCR4$^{hi}$; mature inflammatory monocytes: CD115$^+$ Gr1$^+$ CXCR4$^{lo}$; splenic mature neutrophils: CD115$^-$ Gr1$^{hi}$ CD11b$^{lo}$; splenic inflammatory monocytes: CD115$^+$ Gr1$^+$; CMP: lineage$^-$ cKit$^+$ SCA-1$^-$ CD34$^+$ CD16/32$^{lo}$; GMP: lineage$^-$ cKit$^+$ SCA-1$^-$ CD34$^+$ CD16/32$^{hi}$; MEP: lineage$^-$ cKit$^+$ SCA-1$^-$ CD34$^-$ CD16/32$^-$; and erythrocytes: Ter119$^+$ CD71$^-$ or CD71$^+$. The lineage cocktail was as follows: CD19, B220, CD3e, CD4, Gr1, NK1.1, Ter119, and CD11b.

### Systemic inflammation and cytokine/cytokine receptor blocking in vivo

CFA-induced inflammation was induced by i.p. injection of 200–400 $\mu$l of a 1:1 emulsion of DPBS and CFA (Sigma-Aldrich). Mice were

analyzed at the stated time points after CFA injection. PolyI:C treatments were performed intravenously at 1.4 mg/Kg/d. LT$\beta$R signaling was blocked with 200 $\mu$g of LT$\beta$R-Ig consisting of the LT$\beta$R ectodomain fused with the Fc domain of a mouse IgG1 antibody specific for hen egg lysozyme (Hel-Ig). Control experiments were performed by treatment with 200 $\mu$g HEL-Ig. Inflammatory cytokines TNF-$\alpha$, IFN-$\gamma$, and IL-1$\beta$ were blocked with anti-TNF (#BE0058; Bio-X-Cell), anti-INF-$\gamma$ (#BE0055; Bio-X-Cell), or anti-IL-1$\beta$ (#BE0246; Bio-X-Cell) antibodies at 100 $\mu$g/mouse injected intravenously via the tail vein or retro-orbital sinus immediately before CFA injection. For LT$\beta$R-Ig treatment at steady state, mice were injected with 100 $\mu$g LT$\beta$R-Ig i.v. once a week for 3 wk and analyzed 3 wk after the first injection. In vivo BrdU incorporation was performed by BrdU injection i.v. (1 mg/mouse). Flow cytometry detection of incorporated BrdU was performed with FITC or APC BrdU Flow Kit (BD Biosciences) by following the manufacturer's protocol.

### Infection models

Adult male mice between 8 and 13 wk were used for the infections. Mice were infected via retro-orbital injection. C57BL/6 mice were infected with 50,000 CFU and *Lepr*-cre; *Ltbr*$^{fl/fl}$ mice with 370,000 CFU *L. Monocytogenes*. CFU titers were determined by plating titrated amounts of spleen and liver homogenate on brain heart infusion plates. Briefly, spleen and liver were harvested 3 d after infection and weighed. Tissues were then mashed through a 70-$\mu$m strainer, and titrated dilutions were generated in 1% Triton X-100. The tissue homogenates were then plated on brain heart infusion plates and grown overnight at 37°C. Adult female mice were infected by vaginal inoculation with 1,000 PFU HSV-2. Mice were scored for disease progression every day beginning 4 d after inoculation, as previously described (Iijima et al, 2011). Vaginal swabs were taken from infected mice at day 5 to determine HSV2 PFUs, as described (Iijima et al, 2011).

### MSC sorting and RNA-sequencing

Long bones were flushed with HBSS supplemented with 2% of heat-inactivated fetal bovine serum, penicillin/streptomycin, L-glutamine, Hepes, and 200 U/ml collagenase IV (Worthington Biochemical Corporation), and digested for 30 min at 37°C. Cell clumps were dissociated by gentle pipetting. Cells were filtered through a 100-$\mu$m nylon mesh and washed with HBSS/2% FBS. Hematopoietic cells were depleted by staining with biotin-conjugated CD45 and Ter119 antibodies, and using Dynabeads Biotin Binder (#11047; Invitrogen). After the depletion of hematopoietic cells, the remaining cells were stained with antibodies against CD31, CD144, and PDGFR$\beta$. BM MSCs were identified as CD45$^-$ Ter119$^-$ CD31$^-$ CD144$^-$ PDGFR$\beta^+$ cells. Sorting was performed using a BD FACSAria II. Cells were sorted directly into 350 $\mu$l RLT Plus buffer (QIAGEN), and RNA was extracted using the RNeasy Plus Micro Kit (#74034; QIAGEN). RNA sequencing was performed by the Yale Center for Genome Analysis using the Illumina HiSeq 2000 system, with paired-end 2 × 76 bp read length. The sequencing reads were aligned onto the *Mus musculus* GRCm38/ mm10 reference genome using HISAT2 v2.2.1 software. The mapped reads were converted into the count matrix with default parameters using HTSeq v0.8.0 software, followed by the variance stabilizing

transformation offered by DESeq2. Differentially expressed genes were identified using the same software, DESeq2, based on a negative binomial generalized linear model, and visualized in hierarchically clustered heatmaps using the pheatmap v1.0.12 in R package. Gene set enrichment analyses were conducted using the preranked gene set enrichment algorithm, fgsea v1.20.0, with default parameters on the MSigDB Hallmark gene set. Gene sets were considered significantly enriched when the adjusted *P*-value is less than 0.05.

## Data Availability

Accession number to any data relating to the article was deposited in NCBI (GSE218505).

## Supplementary Information

## Acknowledgements

We thank Mathias Skadow, Harding H Luan, Neeha Kothapali, and Josh Yan for technical assistance. We thank Shuang Yu and Ruslan Medzhitov for help with *Listeria* infections. LT$\beta$R-Ig and HEL-Ig were generously provided by Linda Burkly and Mia Rushe (Biogen Inc., Ma.). These studies were funded by the NIH (RO1AI113040 awarded to JP Pereira); X Feng was funded by the NIH (T32 DK007356). S Zehentmeier was funded by a fellowship from the German Research Foundation (DFG) ZE1060/1-1. AV Tumanov was supported by the Cancer Prevention and Research Institute of Texas (CPRIT) grant (RP220470). J Choi was supported by the National Research Foundation of Korea (NRF) grant funded by the South Korean government (2020R1F1A1076705).

### Author Contributions

VY Lim: conceptualization, data curation, formal analysis, and writing—review and editing.
X Feng: data curation, formal analysis, and writing—review and editing.
R Miao: data curation and formal analysis.
S Zehentmeier: data curation and formal analysis.
N Ewing-Crystal: data curation and formal analysis.
M Lee: formal analysis.
AV Tumanov: resources and writing—review and editing.
JE Oh: data curation, formal analysis, and writing—review and editing.
A Iwasaki: resources and writing—review and editing.
A Wang: conceptualization and writing—review and editing.
J Choi: conceptualization, formal analysis, supervision, methodology, and writing—review and editing.
JP Pereira: conceptualization, data curation, formal analysis, supervision, funding acquisition, project administration, and writing—original draft.

## Conflict of Interest Statement

The authors declare that they have no conflict of interest.

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
