## [Reviewer comments · Life Science Alliance]

Life Science Alliance

Mature B cells and Mesenchymal Stem Cells control emergency myelopoiesis

Vivian Lim, Xing Feng, Runfeng Miao, Sandra Zehentmeier, Nathan Ewing-Crystal, Moonyoung Lee, Alexei Tumanov, Ji Oh, Akiko Iwasaki, Andrew Wang, Jungmin Choi, and Joao Pereira

DOI: <https://doi.org/10.26508/lsa.202301924>

Corresponding author(s): Joao Pereira, Yale University and Jungmin Choi, Yale School of Medicine

Review Timeline:

Submission Date:	2023-01-13
Editorial Decision:	2023-01-18
Revision Received:	2023-01-19
Editorial Decision:	2023-01-20
Revision Received:	2023-01-20
Accepted:	2023-01-23

Transaction Report:

Please note that the manuscript was previously reviewed at another journal and the reports were taken into account in the decision-making process at Life Science Alliance. Since the original reviews are not subject to Life Science Alliance's transparent review process policy, the reports and author response cannot be published.

January 18, 2023

Re: Life Science Alliance manuscript #LSA-2023-01924-T

Dr. Joao P Pereira
Yale University School of Medicine
Immunobiology
300 Cedar Street, S541A
New Haven, CT 6519

Dear Dr. Pereira,

Thank you for submitting your manuscript entitled "Mature B cells and Mesenchymal Stem Cells control emergency myelopoiesis" to Life Science Alliance. We invite you to submit a revised manuscript addressing Reviewer 3's remaining comments.

When submitting the revision, please include a letter addressing the reviewer's comments point by point.

Thank you for this interesting contribution to Life Science Alliance. We are looking forward to receiving your revised manuscript.

Sincerely,

B. MANUSCRIPT ORGANIZATION AND FORMATTING:

January 20, 2023

RE: Life Science Alliance Manuscript #LSA-2023-01924-TR

Dr. Joao P Pereira
Yale University
Immunobiology
300 Cedar Street, S541A
New Haven, CT 6519

Dear Dr. Pereira,

Thank you for submitting your revised manuscript entitled "Mature B cells and Mesenchymal Stem Cells control emergency myelopoiesis". We would be happy to publish your paper in Life Science Alliance pending final revisions necessary to meet our formatting guidelines.

- please add ORCID ID for secondary corresponding author-they should have received instructions on how to do so
- please use the [10 author names, et al.] format in your references (i.e. limit the author names to the first 10)
- please add a figure callout for Figure 4K; Figure S2M; Figure S3L
- GEO dataset GSE218505 should be made publicly accessible at this point
- there should be a Discussion section, for this article you can update the Results section to "Results and Discussion" if you prefer since you summarize your findings at the end of that section

A. FINAL FILES:

B. MANUSCRIPT ORGANIZATION AND FORMATTING:

**Submission of a paper that does not conform to Life Science Alliance guidelines will delay the acceptance of your

manuscript.**

The license to publish form must be signed before your manuscript can be sent to production. A link to the electronic license to publish form will be sent to the corresponding author only. Please take a moment to check your funder requirements.

Sincerely,

January 23, 2023

RE: Life Science Alliance Manuscript #LSA-2023-01924-TRR

Dr. Joao P Pereira
Yale University
Immunobiology
300 Cedar Street, S541A
New Haven, CT 6519

Dear Dr. Pereira,

Thank you for submitting your Research Article entitled "Mature B cells and Mesenchymal Stem Cells control emergency myelopoiesis". It is a pleasure to let you know that your manuscript is now accepted for publication in Life Science Alliance. Congratulations on this interesting work.

DISTRIBUTION OF MATERIALS:

Again, congratulations on a very nice paper. I hope you found the review process to be constructive and are pleased with how the manuscript was handled editorially. We look forward to future exciting submissions from your lab.

Sincerely,
